# Synthesis and Cytotoxic Activity Study of Novel 2-(Aryldiazenyl)-3-methyl-1*H*-benzo[*g*]indole Derivatives

**DOI:** 10.3390/molecules26144240

**Published:** 2021-07-12

**Authors:** Manar M. Arafeh, Ebrahim Saeedian Moghadam, Sirin A. I. Adham, Raphael Stoll, Raid J. Abdel-Jalil

**Affiliations:** 1Faculty of Pharmacy, Middle East University, Amman 11831, Jordan; manararafeh@yahoo.com; 2Chemistry Department, College of Science, Sultan Qaboos University, Muscat 123, Oman; saeedian1985@gmail.com; 3Biology Department, College of Science, Sultan Qaboos University, Muscat 123, Oman; sadham@squ.edu.om; 4Biomolecular NMR, Ruhr University of Bochum, D-44780 Bochum, Germany; raphael.stoll@ruhr-uni-bochum.de

**Keywords:** anti-cancer, synthesis, indole, heterocycles

## Abstract

A novel series of 2-(aryldiazenyl)-3-methyl-1*H*-benzo[*g*]indole derivatives (**3a**–**f**) were prepared through the cyclization of the corresponding arylamidrazones, employing polyphosphoric acid (PPA) as a cyclizing agent. All of the compounds (**3a**–**f**) were characterized using ^1^H NMR, ^13^C NMR, MS, elemental analysis, and melting point techniques. The synthesized compounds were evaluated for cytotoxic activity against diverse human cancer cell lines by the National Cancer Institute. While all of the screened compounds were found to be cytotoxic at a 10 µM concentration, two of them (**2c**) and (**3c**) were subjected to five dose screens and showed a significant cytotoxicity and selectivity.

## 1. Introduction

Cancer is a serious threat to human beings; from the medical viewpoint, it is a complicated complex of genetic diseases, including abnormal and uncontrolled cell growth and proliferation, with the potential to spread throughout the body. Currently, great efforts are being made to prevent or diminish cancer incidence and increase its treatment efficacy, but unfortunately, cancer is still a major cause of morbidity and mortality worldwide and the second most frequent cause of death in the United States.

Combination chemotherapy using antineoplastic agents with a different mechanism of action is one potential approach used to combat cancer [1]. However, the significant drug resistance and narrow dosing window of these drugs, combined with a lack of selectivity, decrease the efficacy of cancer chemotherapy. Thus, incorporating two functional groups in a single molecule, each with a different mechanism of action, might enhance cancer treatment. These limitations in cancer chemotherapy using the currently available drugs encourage drug discovery researchers to find new efficient chemotherapeutic agents for cancer treatment [2]. Nitrogen-containing heterocyclic compounds are a class of compounds that have recently attracted an increased interest within the pharmaceutical community due to their widespread and often diverse biological activities [3]. The indole-based compounds in particular are common building blocks for many medicinally active drugs and are considered as target pharmacophores for the development of therapeutic agents [4,5,6,7,8,9]. During the last decade, various benzo[*g*]indoles have been synthesized and found to exhibit a broad spectrum of biological activities, among which the interchelating anti-tumor activity is of considerable pharmaceutical interest [10]. Certain benzoindoles are useful structures with interesting bioactivities, such as LOX inhibition, bromodomain and extra-terminal (BET) inhibition, anticancer effects, non-covalent Keap1-Nrf2 protein-protein interaction inhibition, and microsomal prostaglandin E2 synthase-1 inhibition [11,12,13,14,15,16,17,18,19,20]. The incorporation of azo moiety as a structural motif has enhanced the cytotoxicity of various systems [21,22]. Ross and Warwick synthesized some azobenzenes which were reduced selectively in vivo to the corresponding diamine and are good alkylating agents [23].

In order to combine the features of two of the most important drug classes in cancer therapy, interchelating and alkylating agents, we report here the synthesis of a new series of 2-(aryldiazenyl)-3-methyl-1*H*-benzo[*g*]indoles and their anti-tumor activities.

## 2. Results and Discussion

### 2.1. Synthesis of Benzo[g]indoles

The general synthetic pathway is shown in Scheme 1. The 1-arylhydrazono-1-chloroacetones (**1**) were prepared from readily available substituted anilines and α-chloroacetylacetone using a standard method [24], which involves the diazotization of the substituted anilines with sodium nitrite in situ, followed by coupling with α-chloroacetylacetone to give the desired 1-arylhydrazono-1-chloroacetones (**1a**–**f**) in a 70–80% yield. The 1-arylhydrazono-1-chloroacetones (**1a**–**f**) were then coupled with 1-naphthylamine to yield the corresponding arylamidrazones (**2a**–**f**), which are then cyclized using polyphosphoric acid (PPA) as a catalyst to produce benzo[*g*]indole target products (**3a**–**f**) in a 50–60% yield.

The synthetic pathway illustrated here has several important advantages. First of all, it represents a fairly straightforward, three-step procedure, which employs readily available starting materials, leading to benzo[*g*]indole derivatives (**3a**–**f**) in good yields. At the same time, a range of derivatives are readily available for each of the starting materials, which subsequently allows for chemical diversification at each step of the synthesis. These considerations particularly apply to the substituted anilines, which are used as part of the first and the second step of the synthesis.

The ^1^H NMR spectra of arylamidrazones (**2a**–**f**) show two doublets at δ (8.1–8.2) ppm and δ (7.9–8.1) ppm assigned to the two protons, H-14 and H-17 (as numbered in Scheme 1), which appear as the most de-shielded protons, with a coupling constant in the range of 6.7–9.0 Hz for H-14 and 6.4–8.2 Hz for H-17. The methyl protons resonate around δ 2.6–2.8 ppm as sharp singlets. The H-9 protons appear as the most upfield protons among the aromatic signals due to the anisotropic effect of the adjacent carbonyl group or imine functional group and appear as doublets at around δ 6.0–6.5 ppm, with a coupling constant of 7.3 Hz. In the ^13^C NMR spectra, the methyl carbons resonate in the range of δ 23.8–24.0 ppm, while the carbonyl carbon resonates at around δ 194.1–194.5 ppm. The carbons in the imine bonds C=N carbons resonate in the range of δ 137.8–147.9 ppm. Aromatic carbons resonate at around δ 108.7–137.2 ppm.

The combined use of 2D homo and hetero-nuclear chemical shift correlation spectroscopy (H/H COSY, C/H COSY, and HMBC) allowed for the unambiguous and complete assignment of the proton and carbon chemical shifts of the target benzo[*g*]indoles (**3a**–**f**). In the ^1^H NMR spectra of (**3a**–**f**), the exchangeable N-H protons resonate in the range of δ 9.5–9.7 ppm and appear as broad singlets. The methyl protons resonate at around δ 2.7–2.8 ppm, and they appear as sharp singlets. The H-10 protons appear in the most de-shielded region among the aromatic protons, and they appear as doublets at around δ 7.9–8.1 ppm, with a coupling constant of 6.4–8.2 Hz. In the ^13^C NMR spectra of the benzoindoles, methyl carbons resonate in the range of 8.6–8.9 ppm, and C-5 resonate at around δ 151.5–165.2 ppm.

### 2.2. Cytotoxic Activity Screening

The benzo[*g*]indoles synthesized in this study were considered for anti-cancer activity screening, as part of the developmental therapeutics program of the National Cancer Institute (NCI) [25,26,27]. After the primary cytotoxic activity screening of all the synthesized compounds, the benzo[*g*]indoles, (**3a**) and (**3c**)**,** were selected for in-depth studies, together with their arylamidrazone precursors, (**2a**) and (**2c**), respectively. As part of these tests, a comprehensive screen against 60 different cancer cell lines was performed, which includes various cell lines representative of non-small cell lung cancer, colon cancer, breast cancer, ovarian cancer, leukemia, renal cancer, melanoma, prostate cancer, and cancer of the central nervous system (CNS).

The results for each compound are reported as the growth percentage (GP) of treated cells in comparison to those untreated control cells. The growth percentage values at a single dose of 10 µM for 60 cancer cell lines are shown in Table 1. Overall, four tested compounds were found to be cytotoxic at µM concentrations (GI_50_ and LC_50_ concentrations of around 10 μM). This toxicity is not unusual, since benzoindoles are known to affect cell survival and proliferation [28]. Interestingly, the screening also revealed that apart from the active benzo[*g*]indoles, (**3a**) and (**3c**), their precursors, i.e., the arylamidrazones (**2a**) and (**2c**), were also cytotoxic, albeit with a different target cell specificity. Furthermore, a rather unexpected activity was found for compounds (**2c**) and (**3c**)**,** as they showed a growth reduction equal to or less than 50% in the initial one-dose analysis. These compounds were selected for further evaluation in the full panel of 60 human tumor cell lines using five different concentrations, and the negative growth present value represents the highest activity of the compound.

A comparison of the values for growth percentage of inhibition of the two compounds, (**2c**) and (**3c**)**,** at 100 µM, GI50, TGI, and LC50 are listed in Table 2. The arylamidrazone (**2c**) showed a considerable activity against two leukemia cancer cells, HL-60(TB) (GI50 equals to 3.9 µM) and MOLT-4 (GI50 equals to 2.3 µM), and also against the melanoma cell line, MALME-3M (GI50 of 5.44 µM), and the breast cancer cell line, HS 578T (GI50 of 6.05 µM). A better inhibition activity was observed for the corresponding benzo[*g*]indole compound, (**3c**)**,** at nano-molar concentrations, which was specifically active against leukemia cell lines: HL-60(TB) (GI50 of 560 nM). Apparently, the arylamidrazone inhibited cell growth considerably more effectively (close to 50% in MOLT-4 cells (GP = −42), when compared to all the other cell lines investigated, pointing toward a better inhibition activity to leukemia cells (Table 2) (Figure 1). Compound (**3c**) was not only active and specific against the leukemia cell lines, but it also showed activity against the melanoma cell line, SK-MEL-5, with a GP = −46 (Table 2) at a 100 µM concentration and a GI50 as low as 429 nM (Figure 2). Not only leukemia or melanoma, but also two renal cancer cell lines are affected by the benzoindole (**3c**) A498 and RXF 393 cell lines, with a negative growth present value and a GI50 of 605 nM and 336 nM, respectively. The most interesting inhibition activity was noticed against the HS 578T breast cancer cell line, derived from a rare, highly malignant mammary carcinosarcoma cell line, possessing histopathologic features of both carcinoma and sarcoma (bone cancer) [29,30]. This HS 578T cell line was inhibited by the maximum growth inhibition, when compared with the GI50s of the other cell lines, and the lowest recorded value by this compound was 53 nM, with a −44 value of growth inhibition (Figure 3). These values represent the lowest concentration of the compound. (**3c**) is therefore a very potent compound not only for leukemia, but also for breast cancer. Interestingly, compounds (**2c**) and (**3c**) contain the nitroaniline moiety, while the unsubstituted aniline compounds, (**2a**) and (**3a**), are less active and do not show a clear ‘trend’ or specificity as far as certain cancer cell lines are concerned.

Nonetheless, as the arylamidrazone (**2c**) illustrates, the simple hydrophobic interaction with DNA strands or metabolic enzymes do not seem to be the only reason for cytotoxicity. Here, the azo group present in the benzo[*g*]indoles, as well as arylamidrazones, may play an additional role. This group is known to be metabolized readily, for instance, by Phase I enzymes in the liver, which leads us to the second possible cause of cytotoxicity. Hepatic azo-reduction is known to result in amine ‘fragments’ of the azo compound affected. Such a reduction is also possible for the compounds studied here, which may subsequently break down into smaller fragments—each of which may exhibit its own cytotoxicity. Interestingly, an azo reduction of the corresponding benzo[*g*]indoles and arylamidrazone may result in different, yet partially related ‘fragments’.

Furthermore, benzene ring hydroxylation may occur, and peroxidases may generate fairly toxic *ortho*-quinones. Regardless of the precise metabolic events associated with such aromatic compounds, it is likely that such processes are cell type-specific, i.e., the generation of metabolites will differ in the kind and extent of metabolites, depending on the cell type [30]. It is therefore possible that leukemia cells are particularly prone to some of the compounds tested due to a specific metabolic ‘activation’ of these compounds in leukemia cells. This notion is, of course, speculative at this point and needs further investigation.

Finally, we also need to emphasize that the arylamidrazones, as precursors of the benzo[*g*]indoles, possess a chemistry of their own, which differs from the one of their corresponding cyclization products. For instance, the arylamidrazone contains a highly reactive ketone, which is used in the cyclization process. This group may also react with biomolecules, leading up to modified proteins, enzymes, DNA, or RNA. The latter may show an impaired function and activity. If, and to what extent, such a chemistry occurs in the living cell is still unclear. Nonetheless, it provides an additional mode of action for the arylamidrazones that is worth considering in the future.

## 3. Materials and Methods

### 3.1. Chemistry

All chemicals, reagents, and solvents were bought from Sigma company and were used without further purification. The ^1^H NMR spectra were recorded on a Bruker Unity 300 or 400 MHz spectrometer. The ^13^C NMR spectra were also recorded at 75 or 100 MHz. The chemical shifts are given in ppm and were referenced with the residual solvent resonances relative to tetramethylsilane (TMS). The mass spectra were obtained on a Finnigan MAT 312 mass spectrometer, connected to a PDO 11/34 (DEC) computer system by electron impact. The elemental analyses were determined using a Perkin Elmer Model 240-C instrument (PerkinElmer, Hopkinton, MA, USA). The melting points were measured on a SMP1 Stuart apparatus and are uncorrected. Flash chromatography was conducted using MERCK silica gel (mesh size 230–400 ASTM). Thin-layer chromatography (TLC) was performed on Macherey-Nagel poly gram Sil G/UV254 silica gel plates, with visualization under UV (254 nm).

#### 3.1.1. General Procedure for the Preparation of the 1-arylhydrazono-1-chloroacetone (**1a**–**f**)

At first, the related aromatic amine (5 mmol) was dissolved in dilute hydrochloric acid (10 mL), and then a solution of sodium nitrite (6 mmol) in water (5 mL) was added dropwise at 5 °C. After completing the addition, the mixture was stirred for an extra 30 min. Then, 3-Chloro-2,4-pentandione (5 mmol) in acetic acid (3 mL) was added to the primary mixture, and the reaction mixture was left to stir at 5 °C for 2 h. After the completion of the reaction, detected by TLC, the produced solid was filtered, washed with cold water, and dried under vacuum overnight. The collected solid 1 was used directly for the next step, without further purification and characterization [24].

#### 3.1.2. General Procedure for the Preparation of 1-(naphthylamino)-1-(arylhydrazono)-2-propanone (**2a**–**f**)

To a stirred solution of 1-arylhydrazono-1-chloroacetone 1 (10 mmol) and 1-naphthylamine (10 mmol) in ethanol (10 mL), triethylamine (TEA) (2 mL) was added at room temperature. The reaction mixture was refluxed for 2 hrs and cooled, and water was added (30 mL). The resulting precipitate was filtered off. The product was recrystallized from hot ethanol to yield the corresponding arylamidrazone 2.

*N*-(Naphthalen-1-yl)-2-oxo-*N*’-phenylpropanehydrazonamide (**2a**)

Yield 62%; m.p. 148–150 °C; ^1^H NMR (250 MHz, CDCl_3_): δ = 2.67 (s, 3 H, CH_3_), 8.10 (d, JH,H = 9.8 Hz, 1 H, H-14), 7.88 (d, JH,H = 7.0 Hz, 1 H, H-17), 7.51–7.62 (m, 3 H, H-11, H-15, H-16), 7.21–7.34 (m, 5 H, H-2, H-2’, H-3, H-3’, H-4), 6.90–7.04 (m, 3 H, H-10, NH, *NH), 6.32 (d, JH,H = 7.32 Hz, 1 H, H-9) ppm. ^13^C NMR (75.5 MHz, CDCl_3_): δ = 23.9 (CH_3_), 143.0 (C=N), 194.3 (C=O), 113.2 (C-9), 113.7 (C-2\C-2’), 121.9 (C-14), 123.0 (C-10), 133.2 (C-3\C-3’), 129.3 (C-17), 125.5, 126.2, 126.4, 128.6, 134.5, 135.3. MS: *m*/*z* (%) = 303 (100) [M]^+^, 288 (10) [M-CH_3_]^+^, 260 (26) [M-CH_3_-CO]^+^, 168 (38) [M-CH_3_-CO-HNAr]^+^, 154 (58) [M-CH_3_-CO-HNAr-N]^+^, 143 (42) [M-CH_3_-CO-HNAr-N-C+H]^+^. C_19_H_17_N_3_O (303.36): calcd. C 75.23, H 5.65, N 13.85; found C 75.36, H 5.53, N 13.98.

*N*’-(4-Chlorophenyl)-*N*-(naphthalen-1-yl)-2-oxopropanehydrazonamide (**2b**)

Yield 73%; m.p. 134–136 °C; ^1^H NMR (300 MHz, CDCl_3_): δ = 2.64 (s, 3 H, CH_3_), 8.12 (d, JH,H = 9.0 Hz, 1 H, H-14), 7.93 (d, JH,H = 7.5 Hz, 1 H, H-17), 7.62 (m, 3 H, H-11, H-15, H-16), 7.40 (br. s, 1 H, NH), 7.35 (dd, JH,H = 6.0 Hz, 1 H, H-10), 7.22 (d, JH,H = 6.0 Hz, 2 H, H-3/H-3’), 7.13 (br. s, 1 H, *NH), 6.97 (d, JH,H = 6.0 Hz, 2 H, H-2/H-2’), 6.35 (d, JH,H = 6.0 Hz, 1 H, H-9) ppm. ^13^C NMR (75.5 MHz, CDCl_3_): δ = 23.9 (CH_3_), 141.7 (C=N), 194.2 (C=O), 113.4 (C-9), 114.8 (C-2\C-2’), 121.7 (C-14), 123.3 (C-10), 129.3 (C-3\C-3’), 128.7 (C-17), 125.4, 126.3, 126.5, 126.6, 132.9, 134.5. MS: *m*/*z* (%) = 337 (73) [M]^+^, 322 (26) [M-CH_3_]^+^, 294 (22) [M-CH_3_-CO]^+^, 168 (58) [M-CH_3_-CO-HNAr]^+^, 154 (100) [M-CH_3_-CO-HNAr-N]^+^, 143 (52) [M-CH_3_-CO-HNAr-N-C+H]^+^. C_19_H_16_ClN_3_O (337.80): calcd. C 67.56, H 4.77, N 12.44; found C 68.05, H 4.80, N 12.68.

*N*-(Naphthalen-1-yl)-*N*’-(4-nitrophenyl)-2-oxopropanehydrazonamide (**2c**)

Yield 64%; m.p. 180–182 °C; ^1^H NMR (300 MHz, CDCl_3_): δ = 2.73 (s, 3 H, CH_3_), 8.11–8.16 (m, 3 H, H-11, H-15, H-16), 7.94 (d, JH,H = 9.3 Hz, 1 H, H-14), 7.61–7.73 (m, 3 H, H-17, H-10, NH), 7.57 (br. s, 1 H, *NH), 7.37 (d, JH,H = 7.2 Hz, 2 H, H-3\H-3’), 6.97 (d, JH,H = 9.3 Hz, 2 H, H-2\H-2’), 6.47 (d, JH,H = 7.5 Hz, 1 H, H-9) ppm. ^13^C NMR (75.5 MHz, CDCl_3_): δ = 24.0 (CH_3_), 147.9 (C=N), 194.3 (C=O), 112.8 (C-9), 114.9 (C-2\C-2’), 132.5 (C-3\C-3’), 124.4 (C-10), 121.7 (C-14), 128.8 (C-17), 125.2, 125.9, 126.5, 126.7, 126.8, 134.5, 137.1. MS: *m*/*z* (%) = 348 (55) [M]^+^, 333 (6) [M-CH_3_]^+^, 305 (10) [M-CH3-CO]^+^, 168 (50) [M-CH_3_-CO-HNAr]^+^, 154 (100) [M-CH_3_-CO-HNAr-N]^+^, 143 (33) [M-CH_3_-CO-HNAr-N-C+H]^+^. C_19_H_16_N_4_O_3_ (348.36): calcd. C 65.51, H 4.63, N 16.08; found C 66.10, H 4.63, N 16.01.

*N*’-(4-Bromophenyl)-*N*-(naphthalen-1-yl)-2-oxopropanehydrazonamide (**2d**)

Yield 63%; m.p. 138–140 °C; ^1^H NMR (400 MHz, CDCl_3_): δ = 2.66 (s, 3 H, CH3), 8.09 (d, JH,H = 8.1 Hz, 1 H, H-14), 7.89 (d, JH,H = 7.6 Hz, 1 H, H-17) 7.54–7.61 (m, 3 H, H-11, H-15, H-16), 7.38 (br. s, 1 H, NH), 7.30–7.34 (m, 3 H, H-10, H-3\H-3’), 7.13 (br. s, 1 H, *NH), 6.89 (d, JH,H = 8.8 Hz, 2 H, H-2\ H-2’), 6.32 (d, JH,H = 7.3 Hz, 1 H, H-9) ppm. ^13^C NMR (100.6 MHz, CDCl_3_): δ = 23.9 (CH_3_), 142.1 (C=N), 194.2 (C=O), 113.8 (C-9), 115.6 (C-2\C-2’), 132.5 (C-3\C-3’), 123.7 (C-10), 129.1 (C-17), 122.1 (C-14), 125.8, 126.4, 126.7, 126.9, 133.3, 134.5, 135.9. MS: *m*/*z* (%) = 381 (58) [M]^+^, 366 (22) [M-CH_3_]^+^, 338 (14) [M-CH_3_-CO]^+^, 168 (62) [M-CH_3_-CO-HNAr]^+^, 154 (100) [M-CH_3_-CO-HNAr-N]^+^, 143 (69) [M-CH_3_-CO-HNAr-N-C+H]^+^. C_19_H_16_BrN_3_O (382.25): calcd. C 59.70, H 4.22, N 10.99; found C 59.53, H 4.17, N 11.02.

*N*-(Naphthalen-1-yl)-2-oxo-*N’*-(*p*-tolyl)propanehydrazonamide (**2e**)

Yield 80%, m.p. 144–146 °C; ^1^H NMR (400 MHz, CDCl_3_): δ = 2.66 (s, 3 H, CH_3_), 2.28 (s, 3 H, *CH_3_), 8.10 (d, JH,H = 8.1 Hz, 1 H, H-14), 7.87 (d, JH,H = 8.8 Hz, 1 H, H-17), 7.50–7.60 (m, 3 H, H-11, H-15, H-16), 7.27–7.31 (m, 2 H, H-10, NH), 7.17 (br. s, 1 H, *NH), 7.05 (d, JH,H = 8.3 Hz, 2 H, H-3\H-3’), 6.93 (d, JH,H = 8.3 Hz, 2 H, H-2\H-2’), 6.28 (d, JH,H = 7.3 Hz, 1 H, H-9) ppm. ^13^C NMR (100.6 MHz, CDCl_3_): δ = 23.8 (CH_3_), 140.8 (C=N), 194.1 (C=O), 113.3 (C-9), 114.1 (C-2\C-2’), 130.2 (C-3\C-3’), 123.2 (C-10), 129.0 (C-17), 122.1 (C-14), 125.9, 126.3, 126.5, 128.6, 131.9, 134.9. MS: *m*/*z* (%) = 317 (100) [M]^+^, 302 (38) [M-CH_3_]^+^, 274 (26) [M-CH_3_-CO]^+^, 168 (30) [M-CH_3_-CO-HNAr]^+^, 154 (52) [M-CH_3_-CO-HNAr-N]^+^, 143 (57) [M-CH_3_-CO-HNAr-N-C+H]^+^. C_20_H_19_N_3_O (317.38): calcd. C 75.69, H 6.03, N 13.24; found C 77.45, H 6.05, N 13.00.

*N*-(Naphthalen-1-yl)-*N*’-(naphthalen-2-yl)-2-oxopropanehydrazonamide (**2f**)

Yield 52%; m.p. 162–164 °C; ^1^H NMR (250 MHz, CDCl_3_): δ = 2.75 (s, 3 H, CH_3_), 8.23 (d, JH,H = 9.0 Hz, 1 H, H-18), 7.05 (d, JH,H = 6.0 Hz, 1 H, H-8), 6.61 (m, 2 H, H-2, H-13), 7.28–7.79 (m, 12 H) ppm. ^13^C NMR (75.5 MHz, CDCl_3_): δ = 24.0 (CH_3_), 137.8 (C=N), 194.5 (C=O), 137.8, 136.5, 134.6, 134.1, 133.0, 128.8, 128.6, 126.8, 126.6, 126.5, 126.3, 125.8, 125.6, 125.5, 123.9, 122.4, 122.1, 121.8, 118.9, 115.2, 108.7 ppm. MS: *m*/*z* (%) = 353 (100) [M]^+^, 338 (36) [M-CH_3_]^+^, 310 (19) [M-CH_3_-CO]^+^, 168 (29) [M-CH_3_-CO-HNAr]^+^, 143 (84) [M-CH_3_-CO-HNAr-N-C+H]^+^. C_23_H_19_N_3_O (353.42): calcd. C 78.16, H 5.42, N 11.89; found C 78.22, H 5.31, N 11.70.

#### 3.1.3. General Procedure for the Synthesis of 2-(Aryldiazenyl)-3-methyl-1*H*-benzo[*g*]indoles (**3a**–**f**)

The corresponding 1-(naphthylamino)-1-(arylhydrazono)-2-propanone 2 (1 mmol) was added to PPA (4 g). The reaction mixture was heated to 80 °C with stirring. Once the exothermic phase subsided, the temperature was increased further to 115 °C to complete the reaction for another 15–20 min. The dark colored solution was then poured onto ice water (50 mL), and the resulting mixture was neutralized with ammonium hydroxide solution (10 mL, 30%). The resulting solution was exhaustively extracted with diethylether (10 mL × 3). The combined organic extracts were dried using sodium sulfate, and the solvent was removed under vaccuum. The crude products were subsequently purified by preparative silica gel thin-layer chromatography, using DCM as the eluent.

3-Methyl-2-(phenyldiazenyl)-1*H*-benzo[*g*]indole (**3a**)

Yield 50%; m.p. 174–176 °C; ^1^H NMR (400 MHz, CDCl_3_): δ = 2.69 (s, 3 H, CH_3_), 9.56 (br. s, NH), 7.98 (d, JH,H = 7.8 Hz, 1 H, H-10), 7.85 (d, JH,H = 7.6 Hz, 2 H, H-2\H-2’), 7.81 (d, JH,H = 7.6 Hz, 1 H, H-14), 7.61 (d, JH,H = 8.6 Hz, 1 H, H-17), 7.39–7.47 (m, 5 H, H-11, H-15, H-16, H-3, H-3’), 7.33 (dd, JH,H = 7.2 Hz, 1 H, H-4) ppm. ^13^C NMR (100.6 MHz, CDCl_3_): δ = 9.1 (CH_3_), 119.8 (C-17), 121.5 (C-10), 122.4 (C-2\C-2’), 129.5 (C-14), 130.2 (C-4), 153.4 (C-5), 121.4, 121.9, 122.7, 125.2, 126.3, 132.1, 133.4, 145.2 ppm. MS: *m*/*z* (%) = 285 (100) [M]^+^, 194 (6) [M-ArN]^+^, 180 (38) [M-ArN_2_]^+^. C_19_H_15_N_3_ (285.34): calcd. C 79.98, H 5.30, N 14.73; found C 79.13, H 5.36, N 14.58.

2-((4-Chlorophenyl)diazenyl)-3-methyl-1*H*-benzo[*g*]indole (**3b**)

Yield 55%; m.p. 181–183 °C; ^1^H NMR (400 MHz, CDCl_3_): δ = 2.74 (s, 3 H, CH_3_), 9.57 (br. s, NH), 8.04 (d, JH,H = 7.8 Hz, 1 H, H-10), 7.88 (d, JH,H = 7.6 Hz, 1 H, H-14), 7.85 (d, JH,H = 8.6 Hz, 2 H, H-2\ H-2’), 7.67 (d, JH,H = 8.9 Hz, 1 H, H-17), 7.44–7.54 (m, 5 H, H-11, H-15, H-16, H-3, H-3’) ppm. ^13^C NMR (100.6 MHz, CDCl_3_): δ = 9.1 (CH3), 119.8 (C-17), 121.4 (C-10), 123.9 (C-2\C-2’), 129.5 (C-14), 151.9 (C-5), 121.6, 121.8, 123.1, 125.3, 126.4, 126.5, 129.7, 132.4, 133.5, 135.7, 145.1 ppm. MS: *m*/*z* (%) = 319 (100) [M]^+^, 194 (5) [M-ArN]^+^, 180 (34) [M-ArN_2_]^+^. C_19_H_14_ClN_3_ (319.79): calcd. C 71.36, H 4.41, N 13.14; found C 71.23, H 4.45, N 12.96.

3-Methyl-2-((4-nitrophenyl)diazenyl)-1*H*-benzo[*g*]indole (**3c**)

Yield 53%; m.p. 182–184 °C. ^1^H NMR (300 MHz, CDCl_3_): δ = 2.81 (s, 3 H, CH_3_), 9.64 (br. s, NH), 8.36 (d, JH,H = 9.2 Hz, 2 H, H-2\ H-2’), 8.11 (d, JH,H = 7.0 Hz, 1 H, H-10), 8.01 (d, JH,H = 9.2 Hz, 2 H, H-3\ H-3’), 7.91 (d, JH,H = 9.5 Hz, 1 H, H-14), 7.70 (d, JH,H = 8.5 Hz, 1 H, H-17), 7.49–7.59 (m, 3 H, H-11, H-15, H-16) ppm. ^13^C NMR (75.5 MHz, CDCl_3_): δ = 8.9 (CH_3_), 119.4 (C-17), 121.2 (C-10), 124.9 (C-2\C-2’), 146.5 (C-5), 121.8, 122.6 126.3, 126.9, 129.3, 135.2 ppm. MS: *m*/*z* (%) = 330 (100) [M]^+^, 194 (9) [M-ArN]^+^, 180 (32) [M-ArN_2_]^+^. C_19_H_14_N_4_O_2_ (330.34): calcd. C 69.08, H 4.27, N 16.96; found C 70.22, H 4.33, N 17.01.

2-((4-Bromophenyl)diazenyl)-3-methyl-1*H*-benzo[*g*]indole (**3d**)

Yield 60%; m.p. 184–186 ℃; ^1^H NMR (400 MHz, CDCl_3_): δ = 2.69 (s, 3 H, CH_3_), 9.54 (br. s, NH), 8.02 (d, JH,H = 7.8 Hz, 1 H, H-10), 7.83 (d, JH,H = 7.6 Hz, 1 H, H-14), 7.73 (d, JH,H = 8.8 Hz, 2 H, H-2\H-2’), 7.63 (d, JH,H = 8.8 Hz, 1 H, H-17), 7.56 (d, JH,H = 8.9 Hz, 2 H, H-3\H-3’), 7.42–7.50 (m, 3 H, H-11, H-15, H-16) ppm. ^13^C NMR (100.6 MHz, CDCl_3_): δ = 9.1 (CH_3_), 119.8 (C-2\C-2’), 121.4 (C-10), 126.5 (C-3\C-3’), 132.7 (C-17), 151.8 (C-5), 121.7, 124.1, 126.4, 129.6 ppm. MS: *m*/*z* (%) = 363 (100) [M]^+^, 194 (9) [M-ArN]^+^, 180 (53) [M-ArN_2_]^+^. C_19_H_14_BrN_3_ (364.24): calcd. C 62.65, H 3.87, N 11.54; found C 63.37, H 3.82, N 11.49.

3-Methyl-2-((*p*-tolyl)diazenyl)-1*H*-benzo[*g*]indole (**3e**)

Yield 51%; m.p. 176–178 °C; ^1^H NMR (400 MHz, CDCl_3_): δ = 2.75 (s, 3 H, CH_3_), 2.43 (s, 3 H, *CH_3_), 9.62 (br. s, NH), 8.05 (d, JH,H = 7.8 Hz, 1 H, H-10), 7.88 (d, JH,H = 7.6 Hz, 1 H, H-14), 7.83 (d, JH,H = 8.3 Hz, 2 H, H-2\H-2’), 7.68 (d, JH,H = 8.6 Hz, 1 H, H-17), 7.46–7.54 (m, 3 H, H-11, H-15, H-16), 7.30 (d, JH,H = 8.3 Hz, 2 H, H-3\H-3’) ppm. ^13^C NMR (100.6 MHz, CDCl_3_): δ = 9.0 (CH_3_), 21.9 (*CH_3_), 119.8 (C-17), 121.3 (C-10), 122.7 (C-2\C-2’), 129.5 (C-14), 130.2 (C-3\C-3’), 151.5 (C-5), 121.4, 121.6, 121.9, 125.2, 126.1, 126.2, 131.9, 133.3, 140.6, 145.2 ppm. MS: *m*/*z* (%) = 299 (100) [M]^+^, 194 (5) [M-ArN]^+^, 180 (21) [M-ArN_2_]^+^. C_20_H_17_N_3_ (299.37): calcd. C 80.24, H 5.72, N 14.04; found C 81.12, H 5.64, N 14.17.

3-Methyl-2-((naphthalen-2-yl)diazenyl)-1*H*-benzo[*g*]indole (**3f**)

Yield 53%; m.p. 174–176 °C; ^1^H NMR (250 MHz, CDCl_3_): δ = 2.76 (s, 3 H, CH_3_), 9.62 (br. s, NH), 8.90 (d, JH,H = 8.2 Hz, 1 H, H-7’), 8.11 (d, JH,H = 7.9 Hz, 1 H, H-10), 7.43–7.92 (m, 11 H) ppm. ^13^C NMR (62.9 MHz, CDCl_3_): δ = 8.8 (CH_3_), 119.5 (C-17), 165.1 (C-5), 112.4, 121.0, 121.3, 123.4, 125.9, 126.0, 126.4, 126.5, 128.1, 129.2, 130.0 ppm. MS: *m*/*z* (%) = 335 (100) [M]^+^, 180 (42) [M-ArN_2_]^+^. C_23_H_17_N_3_ (335.40): calcd. C 82.36, H 5.11, N 12.53; found C 82.25, H 5.15, N 12.3.

### 3.2. In Vitro Cytotoxicity Assay

The cellular response to drugs was evaluated using the sulforhodamine B assay, as described in literature [31]. Briefly, the human tumor cell lines making up the NCI cancer screening panel were routinely grown in an RPMI 1640 medium containing 5% fetal bovine serum and 2 mM L-glutamine. The cells were inoculated into 96-well microtiter plates in 100 mL of the complete medium at densities ranging from 5000 to 40,000 cells/well. The microtiter plates containing cells were incubated for 24 h prior to the addition of experimental drugs. Following the addition of the drugs, the plates were incubated for an additional 48h, and the cells were fixed with TCA, washed, and stained with sulforhodamine B (Sigma Chemical Co., St. Louis, MO, USA) at 0.4% (*w*/*v*) in 1% acetic acid. After washing with 1% acetic acid, the stain was solubilized with 10 mM unbuffered Tris base, and the absorbance was measured on a Bio-Tek microplate reader. Dose-response parameters were calculated, as reported in the literature [32].

## 4. Conclusions

In summary, we successfully prepared a new series of benzo[*g*]indoles (**3a**–**f**) following a simple, yet effective synthetic avenue, which provides several opportunities for the chemical diversification of the end products. All of the compounds (**3a**–**f**) were characterized using spectroscopic methods. The initial in vitro cancer cell screen revealed a rather interesting cytotoxicity of some of these compounds against a few cancer cell lines, including leukemia, melanoma, renal, and breast cancer cell lines. Further in vivo experiments will be conducted in the future.

## Data Availability

All of the data is included in the manuscript.

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
