# Peer review of "Synthesis and Cytotoxic Activity Study of Novel 2-(Aryldiazenyl)-3-methyl-1H-benzo[g]indole Derivatives"

_molecules, 2021, doi:10.3390/molecules26144240_

Round 1
Reviewer 1 Report
In the present manuscript, Abdel-Jalil and coworkers describe a novel synthesis of 2-(aryldiazenyl)-[1H]-3-methylbenzo[g]indole derivatives and an investigation of their cytotoxic activity. The synthetic approach is short and straightforward. The products have been obtained in reasonable yields and all new compounds were satisfactorily characterized based on their corresponding spectroscopic data as shown in the experimental part. In the study of the biological activity, it was found that all screened compounds exhibited cytotoxic activities and the derivatives 2c and 3c were found to be significantly active. Generally, the manuscript has been written well, however, some aspects summarized below require a revision:
- On page 1, lines 26–29, the authors say: "The indole-based compounds, in particular, are common building blocks for many medicinally active drugs and considered as target pharmacophores for the development of therapeutic agents [4]." In the corresponding references, several seminal and basic reviews on the field of "indole-based compounds" emphasizing the statement given in this sentence should have been cited (this is missing completely!), for example: J. Chem. Soc., Perkin Trans. 1, 2000, 1045–1075; Pure Appl. Chem. 2003, 75, 1417–1432; Curr. Org. Chem. 2005, 9, 1601–1614; Chem. Rev. 2006, 106, 2875–2922; Chem. Sci. 2013, 4, 29–41.
- Page 1, line 29: "benzo[g]indoles", the "g" must be in italic. This should be corrected throughout the whole text, including the compound names given in section 3.1.!
- Page 1, lines 33–34: "non-covalent", "protein-protein interaction inhibition", and "microsomal prostaglandin E2 synthase". Erroneously, several of these words have been written starting with a capital first letter.
- Page 1, line 41: "[1H]-3-methylbenzo[g]indoles", the "H" and the "g" must be in italic. This should be corrected throughout the whole text, including the compound names given in section 3.1.!
- Page 3, line 96, it should say: "Results for each compounds are reported as ..."
After the implementation of these additions and corrections, this very nice manuscript is strongly recommended for publication in Molecules.
Author Response
Thank you very much for your valuable comments. We tried to apply all the comments and do corrections based on your advice. Here there are answers to your comments:
- All suggested references were added inside the manuscript in a suitable place.
- The benzo[g]indoles phrase was corrected throughout the manuscript.
- All mentioned words were corrected.
- The [1H]-3-methylbenzo[g]indoles phrase were corrected throughout the manuscript.
- The suggested sentence applied exactly in the manuscript.
Reviewer 2 Report
The authors of this paper have prepared a series of 2-diazoaryl-benzoindoles which have subsequently tested for their cytotoxic activity. My opinion is that the content of this manuscript is interesting enough to be published in Molecules providing that the following changes are introduced in the draft:
- The content of Scheme 1 is poorly illustrative. Chemical yields for each transformation 1-3 should be inserted as a Table and the reaction conditions better detailed in Scheme 1.
- In the Experimental Section, the procedure and characterization data for compounds 1 should be given.
- Characterization data presentation is a bit inaccurate. Subscripts are not present and the Greek letter delta for chemical shifts is substituted by another symbol.
Author Response
Thank you very much for your valuable comments. We tried to apply all the comments and do corrections based on your advice. Here there are answers to your comments:
- As suggested, the content of scheme 1 totally changed to present the data in a better way.
- As suggested, the procedure for the preparation of compound 1(a-f) was reported inside the manuscript. In the case of characterization of compound 1(a-f), as those compounds are known and reported in the literature, and also are very irritative and work with these compounds is really hard, we used all of them directly for the next step.
- Characterization data presentation was corrected and all mentioned items were corrected.